# How can classical multidimensional scaling go wrong?

**Rishi Sonthalia**
University of Michigan
rsonthal@umich.edu

**Gregory Van Buskirk**
University of Texas - Dallas
greg.vanbuskirk@utdallas.edu

**Benjamin Raichel**
University of Texas - Dallas
Benjamin.Raichel@utdallas.edu

**Anna C. Gilbert**
Yale University
anna.gilbert@yale.edu

## Abstract

Given a matrix $D$ describing the pairwise dissimilarities of a data set, a common task is to embed the data points into Euclidean space. The classical multidimensional scaling (cMDS) algorithm is a widespread method to do this. However, theoretical analysis of the robustness of the algorithm and an in-depth analysis of its performance on non-Euclidean metrics is lacking.

In this paper, we derive a formula, based on the eigenvalues of a matrix obtained from $D$, for the Frobenius norm of the difference between $D$ and the metric $D_{\mathrm{cmds}}$ returned by cMDS. This error analysis leads us to the conclusion that when the derived matrix has a significant number of negative eigenvalues, then $\|D - D_{\mathrm{cmds}}\|_F$, after initially decreasing, will eventually increase as we increase the dimension. Hence, counterintuitively, the quality of the embedding degrades as we increase the dimension. We empirically verify that the Frobenius norm increases as we increase the dimension for a variety of non-Euclidean metrics. We also show on several benchmark datasets that this degradation in the embedding results in the classification accuracy of both simple (e.g., 1-nearest neighbor) and complex (e.g., multi-layer neural nets) classifiers decreasing as we increase the embedding dimension.

Finally, our analysis leads us to a new efficiently computable algorithm that returns a matrix $D_l$ that is at least as close to the original distances as $D_t$ (the Euclidean metric closest in $\ell_2$ distance). While $D_l$ is not metric, when given as input to cMDS instead of $D$, it empirically results in solutions whose distance to $D$ does not increase when we increase the dimension and the classification accuracy degrades less than the cMDS solution.

## 1   Introduction

Multidimensional scaling (MDS) refers to a class of techniques for embedding data into Euclidean space given pairwise dissimilarities [Carroll and Arabie, 1998, Borg and Groenen, 2005]. Apart from the general usefulness of dimensionality reduction, MDS has been used in a wide variety of applications including data visualization, data preprocessing, network analysis, bioinformatics, and data exploration. Due to its long history and being well studied, MDS has many variations such as non-metric MDS [Shepard, 1962a,b], multi-way MDS [Kroonenberg, 2008], multi-view MDS [Bai et al., 2017], confirmatory or constrained MDS [Heiser and Meulman, 1983], etc. (See France and Carroll [2010], Cox and Cox [2008] for surveys).

The basic MDS formulation involves minimizing an objective function over a space of embeddings. There are two main objective functions associated with MDS: STRESS and STRAIN.

35th Conference on Neural Information Processing Systems (NeurIPS 2021).

The STRAIN objective (Equation 1 below) was introduced by Torgerson [1952], whose algorithm to solve for this objective is now commonly referred to as the classical MDS algorithm (cMDS).

$$X_{\text{cmds}} := \arg\min_{X \in \mathbb{R}^{r \times n}} \left\| X^T X - \left( \frac{-VDV}{2} \right) \right\|. \tag{1}$$

Here $V$ is the centering matrix given by $V := I - \frac{1}{n}J$, and $I$ is the identity matrix and $J$ is the matrix of all ones. cMDS first centers the squares of the given distance matrix and then uses its spectral decomposition to extract the low dimensional embedding. cMDS is one of the oldest and most popular methods for MDS, and its popularity is in part due to the fact that this decomposition is fast and can scale to large matrices. The point set produced by cMDS, however, is not necessarily the point set whose Euclidean distance matrix is closest under say Frobenius norm to the input dissimilarity matrix. This type of objective is instead captured by STRESS, which comes in a variety of related forms. In particular, in this paper we consider the SSTRESS objective (see Equation 2).

Specifically, given an embedding $X$, let $EDM(X)$ be the corresponding Euclidean distance matrix, that is $\text{EDM}(X)_{ij} = \|X_i - X_j\|_F^2$, where $X_i, X_j$ are the $i$th and $j$th columns of $X$. If $D$ is a dissimilarity matrix whose entries are squared, then we are interested in the matrix,

$$D_t := \arg\min_{D' = EDM(X), X \in \mathbb{R}^{r \times n}} \|D' - D\|_F^2. \tag{2}$$

Note that the reason we assume our dissimilarity matrix has squared entries is because the standard EDM characterizations uses squared entries (see further discussion of EDMs below). Equation 2 is a well studied objective [Takane et al., 1977, Hayden and Wells, 1988, Qi and Yuan, 2014].

There are a number of similarly defined objectives. If one considers this objective when the matrix entries are not squared (i.e. $\sqrt{D'_{ij}} - \sqrt{D_{ij}}$), then it is referred to as STRESS. If one further normalizes each entry of the matrix difference by the input distance value (i.e. $(\sqrt{D'_{ij}} - \sqrt{D_{ij}})/\sqrt{D_{ij}}$) then it is called Sammon Stress. In this paper, we are less concerned with the differences between different types of Stress, and instead focus on how the cMDS solution behaves generally under a Stress type objective. Thus for simplicity we focus on SSTRESS. It is important to note that there are algorithms to solve the SSTRESS objective, but the main drawback is that they are slow in comparison to cMDS [Takane et al., 1977, Hayden and Wells, 1988, Qi and Yuan, 2014]. Thus, many practitioners default to using cMDS and do not optimize for SSTRESS.

In this paper, we shed light on the theoretical and practical differences between optimizing for these two objectives. Let $D_{\text{cmds}} := \text{EDM}(X_{\text{cmds}})$, where $X_{\text{cmds}}$ is the solution to Equation 1, and let $D_t$ be the solution to Equation 2. We are interested in understanding the quantity

$$\text{err} := \|D - D_{\text{cmds}}\|_F^2. \tag{3}$$

Doing so will provide practitioners with multiple advantages and will guide the development of better algorithms. In particular,

1. Understanding err is the first step in rigorously quantifying the robustness of the cMDS algorithm.
2. If err is guaranteed to be small, then we can use the cMDS algorithm without having to worry about loss in quality of the solution.
3. If err is big, we can make an informed decision about the benefits of the speed of the cMDS algorithm versus the quality of the solution.
4. Understanding when err is big helps us design algorithms to approximate $D_t$ that perform better when cMDS fails.

**Contributions.** Our main theorem, Theorem 1, decomposes err into three components. This decomposition gives insight into when and why cMDS can fail with respect to the SSTRESS objective. In particular, for Euclidean inputs, err naturally decreases as the embedding dimension increases. For non-Euclidean inputs, however, our decomposition shows that after an initial decrease, counterintuitively err can actually increase as the embedding dimension increases. In practice one may not know a priori what dimension to embed into, though one might assume it suffices to embed into some sufficiently large dimension. Importantly, these results demonstrate that when using cMDS to embed, choosing a dimension too large can actually increase error.

This degradation of the cMDS solution is of particular concern in relation to the robustness in the presence of noisy or missing data, as may often be the case for real world data. Several authors [Cayton and Dasgupta, 2006, Mandanas and Kotropoulos, 2016, Forero and Giannakis, 2012] have proposed variations to specifically address robustness with cMDS. However, our decomposition of err, suggests a novel approach. Specifically, by attempting to directly correct for the problematic term in our decomposition (which resulted in err increasing with dimension) we produce a new lower bound solution. We show empirically that this lower bound corrects for err increasing, both by itself and when used as a seed for cMDS. Crucially the running time of our new approach is comparable to cMDS, rather than the prohibitively expensive optimal SSTRES solution. Finally, and perhaps more importantly, we show that if we add noise or missing entries to real world data sets, then our new solution outperforms cMDS in terms of the downstream task of classification accuracy, under various classifiers. Moreover, our decomposition can be used to quickly predict the dimension where the increase in err might occur.

The main contributions of our paper are as follows.

1. We decompose the error in Equation 3 into three terms that depend on the eigenvalues of a matrix obtained from $D$. Using this analysis, we show that there is a term that tells us that as we increase the dimension that we embed into, eventually, the error starts increasing.
2. We verify, using classification as a downstream task, that this increase in the error for cMDS results in the degradation of the quality of the embedding, as demonstrated by the classification accuracy decreasing.
3. Using this analysis, we provide an efficiently computable algorithm that returns a matrix $D_l$ such that if $D_t$ is the solution to Equation 2, then $\|D_l - D\|_F \leq \|D_t - D\|_F$, and empirically we see that $\|D_l - D_t\|_F \leq \|D_{\text{cmds}} - D_t\|_F$.
4. While $D_l$ is not metric, when given as input to cMDS instead of $D$, it results in solutions that are empirically better than the cMDS solution. In particular, this modified procedure results in a more natural decreasing of the error as the dimension increases and has better classification accuracy.

## 2 Preliminaries and Background

In this section, we lay out the preliminary definitions and necessary key structural characterizations of Euclidean distance matrices.

**cMDS algorithm.** For completeness, we include the classical multidimensional scaling algorithm in Algorithm 1.

---

**Algorithm 1** Classical Multidimensional Scaling.

---

1: **function** CMDS($D$, $r$)
2:     $X = -V * D * V/2$
3:     Compute $\mu_1 \geq \ldots \geq \mu_r > 0$, $U$ as the eigenvalues and eigenvectors of $X$
4:     return $U * \text{diag}(\sqrt{\mu_1}, \ldots, \sqrt{\mu_r}, 0, \ldots, 0)$

---

**EDM Matrices**

**Definition 1.** *$D \in \mathbb{R}^{n \times n}$ is a Euclidean Distance Matrix (EDM) if and only if there exists a $d \in \mathbb{N}$ such that there are points $x_1, \ldots, x_n \in \mathbb{R}^d$ with*

$$D_{ij} = \|x_i - x_j\|_2^2.$$

*Note that unlike other distance matrices, an EDM consists of the squares of the (Euclidean) distances between the points.*

This form permits several important structural characterizations of the cone of EDM matrices.

1. Gower [1985], Schoenberg [1935] show that a symmetric matrix $D$ is an EDM if only if

$$F := -(I - \mathbf{1}s^T)D(I - s\mathbf{1}^T)$$

is a positive semi-definite matrix for all $s$ such that $\mathbf{1}^T s = 1$ and $Ds \neq 0$.

2. Schoenberg [1938] showed that $D$ is an EDM if and only if $\exp(-\lambda \mathrm{D})$ is a PSD matrix with 1s along the diagonal for all $\lambda > 0$. Note here $\exp$ is element wise exponentiation of the matrix.

3. Another characterization is given by Hayden and Wells [1988] in which $D$ is an EDM if and only if $D$ is symmetric, has 0s on the diagonal (i.e., the matrix is hollow), and $\hat{D}$ is negative semi-definite, where $\hat{D}$ is defined as follows

$$QDQ = \begin{bmatrix} \hat{D} & f \\ f^T & \xi \end{bmatrix}. \tag{4}$$

Here $f$ is a vector and

$$Q = I - \frac{2}{v^T v} vv^T \text{ for } v = [1, \ldots, 1, 1 + \sqrt{n}]^T. \tag{5}$$

Note $Q$ is Householder reflector matrix (in particular, it's unitary) and it reflects a vector about the hyperplane $\mathrm{span}(v)^\perp$.

The characterization from Hayden and Wells [1988] is the main characterization that we shall use. Hence we establish some important notation.

**Definition 2.** *Given any symmetric matrix $A \in \mathbb{R}^{n \times n}$, let us define $\hat{A} \in R^{n-1 \times n-1}, f(A) \in \mathbb{R}^{n-1}$, and $\xi(A) \in \mathbb{R}$ as follows*

$$QAQ = \begin{bmatrix} \hat{A} & f(A) \\ f(A)^T & \xi(A) \end{bmatrix}.$$

In addition to characterizations of the EDM cone, we are also interested in the dimension of the EDM.

**Definition 3.** *Given an EDM $D$, the dimensionality of $D$ is the smallest dimension $d$, such that there exist points $x_1, \ldots, x_n \in \mathbb{R}^d$ with $D_{ij} = \|x_i - x_j\|_2^2$.*

*Let $\mathcal{E}(r)$ be the set of EDM matrices whose dimensionality is at most $r$.*

Using Hayden and Wells [1988]'s characterization of EDMs, Qi and Yuan [2014] show $D \in \mathcal{E}(r)$ if and only if $D$ is symmetric, hollow (i.e., 0s along the main diagonal), and $\hat{D}$ in Equation 4 is negative semi-definite with rank at most $r$.

**Conjugation matrices: $Q$ and $V$.** Conjugating distance matrices either by $Q$ (in Equation 5) or by the centering matrix $V$ is an important component of understanding both EDMs and the cMDS algorithm. We observe that $V$ is also (essentially) a Householder matrix like $Q$. Let $w = [1, \ldots, 1]^T$ and observe that $J = ww^T$ so that $V = I - \frac{1}{w^T w} ww^T$.

Qi and Yuan [2014] establishes an important connection between $Q$ and $V$ that we make use of in our analysis in Section 3. Specifically, for any symmetric matrix $A$, we have that

$$VAV = Q \begin{bmatrix} \hat{A} & 0 \\ 0 & 0 \end{bmatrix} Q. \tag{6}$$

Here $\hat{A}$ is the matrix given by Definition 2. This connection gives a new framework in which we can understand the cMDS algorithm. We know that given $D$, cMDS first computes $-VDV/2$. Using Equation 6, this is equal to $-Q \begin{bmatrix} \hat{D} & 0 \\ 0 & 0 \end{bmatrix} Q/2$. Thus, when cMDS computes the spectral decomposition of $-VDV/2$, this is equivalent of computing the spectral decomposition of $\hat{D}$. Then using the characterization of $\mathcal{E}(r)$ from Qi and Yuan [2014], setting all but the largest $r$ eigenvalues to 0 might seem like the optimal solution. However, this procedure ignores condition that the matrix must be hollow. As we shall show, this results in sub-optimal solutions.

## 3 Theoretical Results

Throughout this section we fix the following notation. Let $D$ be a distance matrix with squared entries. Let $r$ be the dimension into which we are embedding. Let $\lambda_1 \leq \ldots \leq \lambda_{n-1}$ be the eigenvalues of $\hat{D}$ and $U$ be the eigenvectors. Let $\boldsymbol{\lambda}$ be an $n$-dimensional vector where $\boldsymbol{\lambda}_i = \lambda_i \mathbb{1}_{\lambda_i > 0 \text{ or } i > r}$ for

$i = 1, \ldots, n-1$ and $\boldsymbol{\lambda}_n = 0$. Let $S = Q * \begin{bmatrix} U & 0 \\ 0 & 1 \end{bmatrix}$. Let $D_t$ be the solution to Problem 2 and $D_{\text{cmds}}$ the resulting EDM from the solution to Problem 1. Let

$$C_1 = \sum_{i=1}^{n-1} \boldsymbol{\lambda}_i^2, \quad C_2 = -\sum_{i=1}^{n-1} \boldsymbol{\lambda}_i, \quad C_3 = \frac{n\|(S \circ S)\boldsymbol{\lambda}\|_F^2 - C_2^2}{2}.$$

Then the main result of the paper is the following spectral decomposition of the SSTRESS error.

**Theorem 1.** *If $D$ is a symmetric, hollow matrix then, $\|D_{\text{cmds}} - D\|_F^2 = C_1 + C_2^2 + C_3$.*

The idea behind the proof of Theorem 1 is to decompose

$$\|D_{\text{cmds}} - D\|_F^2 = \|QD_{\text{cmds}}Q - QDQ\|_F^2$$
$$= 4\|\hat{D}/2 - \hat{D}_{\text{cmds}}/2\|_F^2 + (\xi(D) - \xi(D_{\text{cmds}}))^2 + 2\|f(D) - f(D_{\text{cmds}})\|_F^2,$$

which follows from Definition 2. We relate each of these three terms to $C_1$, $C_2$, and $C_3$. The following lemmas will work towards expressing each of these terms separately. In the following discussion, let $Y_r := X_{\text{cmds}}{}^T X_{\text{cmds}}$ and recall that $X_{\text{cmds}}$ is the solution to the classical MDS problem given in Equation 1. The proofs for the following lemmas are in the appendix.

**Lemma 1.** *If $G$ is a positive semi-definite Gram matrix, then $-\frac{1}{2}V\,EDM(G)\,V = Q \begin{bmatrix} \hat{G} & 0 \\ 0 & 0 \end{bmatrix} Q$*

**Lemma 2.** *The value of the objective function obtained by $X_{\text{cmds}}$ in Equation 1 is $\frac{C_1}{4}$. Specifically, we have that $4\|Y_r - (-VDV)/2\|_F^2 = \sum_{i=1}^{n-1} \boldsymbol{\lambda}_i^2 =: C_1$.*

**Lemma 3.** *$-\frac{1}{2}\hat{D}_{\text{cmds}} = \hat{Y}_r$.*

**Lemma 4.** *If $Tr(D) = 0$, then $(\xi(D) - \xi(D_{\text{cmds}}))^2 = \left(\sum_{i=1}^{n-1} \boldsymbol{\lambda_i}\right)^2 =: C_2^2$.*

**Lemma 5.** *If $D$ is hollow, then $2\|f(D) - f(D_{\text{cmds}})\|_F^2 = \dfrac{n\|(S \circ S)\boldsymbol{\lambda}\|_F^2 - C_2^2}{2} =: C_3$.*

The first term in the error is $C_1$. From the definition, we can see $C_1$ is the sum of the squares of the eigenvalues of $\hat{D}$ corresponding to eigenvectors that are not used (or are discarded) in the cMDS embedding. As we increase the embedding dimension, we use more eigenvectors. Hence as the embedding dimension increases, we see that $C_1$ monotonically decreases. In the case when $D$ is an EDM, we can use all of the eigenvectors so this term will go to zero. On the other hand, if $D$ is not an EDM and $\hat{D}$ has positive eigenvalues (i.e. negative eigenvalues of $-VDV$), then these are eigenvalues that correspond to eigenvectors that cannot be used. Thus, cMDS will always exhibit this phenomenon for $C_1$ regardless of the input $D$ (for both STRAIN and SSTRESS).

The second term is $C_2^2$. This term looks similar to $C_1$, but instead of summing the eigenvalues squared, we first sum the eigenvalues and then take the square. This subtle difference has a big impact. First, we note that as $r$ increases, $C_2$ becomes more negative. Suppose that $D$ is not an EDM, (i.e., $\hat{D}$ has positive eigenvalues) and let $K$ be the number of negative eigenvalues of $\hat{D}$. Then, since $D$ has trace 0, when $r = K$, $C_2$ is negative. Hence, $C_2$ decreases and is eventually negative as $r$ increases which implies that as $r$ increases, there is a a certain value of $r$ after which $C_2^2$ **increases. This results in the quality of the embedding decreasing. As we will see, this term will be the dominant term in the total error.**

While $C_1$ and $C_2$ are simple to understand, $C_3$ is more obtuse. To simplify it, we consider the following. If $\delta$ is an entry of a random $n$ by $n$ unitary matrix, then as $n$ goes to infinity the distribution for $\delta\sqrt{n}$ converges to $\mathcal{N}(0,1)$ and the total variation between the two distributions is bounded above by $8/(n-2)$ [Easton, 1989, Diaconis and Freedman, 1987]. Therefore, we can assume that the variance of an entry of a random $n$ by $n$ orthogonal matrix is about $1/n$. So, heuristically, $S \circ S \approx \frac{1}{n}11^T$ and

$$n\|(S \circ S)\vec{\lambda}\|_F^2 \approx \frac{n}{n^2}\|11^T\boldsymbol{\lambda}\|_F^2 = \frac{n}{n^2}\|1C_2\|_F^2 = C_2^2.$$

Then since $C_3 = \frac{n\|(S \circ S)\boldsymbol{\lambda}\|_F^2 - C_2^2}{2} \approx \frac{C_2^2 - C_2^2}{2}$, we see that, at least heuristically, the overall behavior of $C_3$ is dominated by $C_2$.

---

**Algorithm 2** Lower Bound Algorithm.

---

1: **function** LOWER($D, r$)
2:     Compute $\hat{D}$, $f(D)$ and $\xi(D)$.
3:     Compute $\lambda_1 \leq \ldots \leq \lambda_{n-1}$, $U$ as the eigenvalues and eigenvectors of $\hat{D}$
4:     Initialize $c_i = \lambda_i$ and $c_n = \xi(D)$
5:     Let negative_C$_2$ = 0.
6:     **for** $i = 1 \ldots n - 1$ **do**
7:         **if** $\lambda_i > 0$ or $i > r$ **then**
8:             negative_C$_2$ += $\lambda_i$
9:             Set $c_i$ to 0
10:     $E$ := number of $c$'s not equal to 0
11:     sub := negative_C$_2$/E
12:     **for** $i = 1 \ldots n - 1$ **do**
13:         **if** $c_{n-i}! = 0$ **then**
14:             **if** $c_{n-i} + \text{sub} \leq 0$ **then**
15:                 $c_{n-i}$ += sub, $E$ −= 1, negative_C$_2$ −= sub
16:             **else**
17:                 $E$ −= 1, negative_C$_2$ += $c_{n-1}$, sub = negative_C$_2$/E
18:                 $c_{n-1} = 0$
19:     $c_n$ += sub, negative_C$_2$ −= sub
20:     Let $\hat{D} = U * \text{diag}(c_1, \ldots, c_{n-1}) * U^T$
21:     return $Q * \begin{bmatrix} \hat{D} & f(D) \\ f(D)^T & c_n \end{bmatrix} * Q$

---

From the previous discussion, we see that the $C_2^2$ term in the decomposition is the most vexing due to the term $(\xi(D) - \xi(D_{\text{cmds}}))^2$. This term is from the excess in the trace; that is, the result of discarding the eigenvalues changes the value of the trace from 0 to non-zero. From Lemma 3, we see that cMDS projects $-\hat{D}/2$ onto the cone of PSD matrices (which do not necessarily have trace 0). To retain the trace 0 condition, we perform the following adaptation. Let $\kappa(r)$ be the space of symmetric, trace 0 matrices $A$, such that $\hat{A}$ is negative semi-definite of rank at most $r$ and project $D$ onto $\kappa(r)$ instead. That is, we seek a solution the following problem.

$$D_l := \underset{A \in \kappa(r)}{\arg\min} \|A - D\|_F^2. \tag{7}$$

Clearly this problem is a strict relaxation of the SSTRESS MDS problem and hence we get that

$$\|D - D_t\|_F \geq \|D_l - D\|_F$$

Because we no longer require the solution to be hollow, conjugating by $S$ permits us to rewrite Problem 7 as:

$$\boldsymbol{c} := \underset{\substack{\boldsymbol{c} \in \mathbb{R}^n, \boldsymbol{c}^T 1 = 0, \\ \forall n > i, \, c_i \leq 0, \\ \forall n > j > r, \, c_j = 0}}{\arg\min} \sum_{i=1}^{n-1} (c_i - \lambda_i)^2 + (c_n - \xi(D))^2 \tag{8}$$

**Theorem 2.** *If $\boldsymbol{c}$ is the solution to Problem 8 and $D_l$ is the solution to problem 7 and if we let $M$ be the $n - 1$ by $n - 1$ diagonal matrix with the first $n - 1$ terms of $\boldsymbol{c}$ on the diagonal, then* $S^T D_l S = \begin{bmatrix} M & f(D) \\ f(D)^T & c_n \end{bmatrix}$.

Unlike Problem 2, Problem 8 can be solved directly. We do so via Algorithm 2. We see that in the first loop (lines 6-9), we set $c_i$ to be 0 if $\lambda_i > 0$ or $i > r$ to ensure the proper constraints on $c_i$s (i.e., the eigenvalues). Doing so, we incur a cost of $C_1$. That is, we sum the squares of those eigenvalues. Now the solution after line 9 does not necessarily satisfy $\boldsymbol{c}^T 1 = 0$. In fact, at this stage of the algorithm, we have that $\boldsymbol{c}^T 1 = C_2$, and we have $E$ many $c$'s that are non zero that we can

adjust. That is, we modify these entries on average by $C_2/E$. Because we use a Frobenius norm to measure error, this incurs a cost of $C_2^2 E / E^2 = C_2^2 / E$. Finally, we note that the major computational step of the algorithm is the spectral decomposition. Hence it has a comparable run time to cMDS.

**Theorem 3.** *If $D$ is a symmetric, hollow matrix, then Algorithm 2 computes $D_l$ the solution to 7 in $O(n^3)$ time.*

**Proposition 1.** *If $D$ is a symmetric, hollow matrix, then*

$$\|D_l - D\|_F^2 \geq C_1 + \frac{C_2^2}{r+1}$$

It is important to note that $D_l$ is not a metric. However, if we want an EDM and hence an embedding, we can use $D_l$ as the input to cMDS algorithm. Noting that $\hat{D}_l$ is negative semi-definite matrix of rank at most $r$, we see that if $D_{\text{lcmds}}$ is the metric returned by cMDS on input $D_l$, then we have that $\|D_l - D_{\text{lcmds}}\|_F^2 = 2\|f(D_l) - f(D_{\text{lcmds}})\|_F^2 = 2\|f(D) - f(D_{\text{lcmds}})\|_F^2$. Similar to Lemma 5, we get the following result.

**Lemma 6.** $2\|f(D) - f(D_{\text{lcmds}})\|_F^2 = \dfrac{n\|(S \circ S)\boldsymbol{c}\|_F^2 - \frac{C_2^2}{r+1}}{2} =: C_4.$

Using Lemma 6, we see that

$$\|D_{\text{lcmds}} - D\|_F^2 \leq \|D_{\text{lcmds}} - D_l\|_F^2 + \|D_l - D\|_F^2$$
$$\leq 2\|f(D_{\text{lcmds}}) - f(D_l)\|_F^2 + \|D_t - D\|_F^2.$$
$$\Rightarrow \|D_{\text{lcmds}} - D\|_F^2 - \|D_t - D\|_F^2 \leq \frac{n\|(S \circ S)\boldsymbol{c}\|_F^2 - \frac{C_2^2}{r+1}}{2} = C_4$$

On the other hand, we upper bound $\|D_{\text{cmds}} - D\|_F^2 - \|D_t - D\|_F^2$ using Proposition 1. Taking the difference, we get

$$\|D_{\text{cmds}} - D\|_F^2 - \|D_t - D\|_F^2 \leq \frac{r}{r+1}C_2^2 + C_3$$

Using our heuristics for $C_3$ and $C_4$, we expect $D_{\text{cmds}}$ to be be much further from $D_t$, than $D_{\text{lcmds}}$. These claims are experimentally verified in the next section.

**Solving Equation 2.** Another approach would be to solve Equation 2 directly. However, solving this problem is much more computationally expensive and works such as Qi and Yuan [2014] present new algorithms to do so. Algorithm 2 can be used to compute the solution as well. Here we Dykstra's method of alternating projections as Hayden and Wells [1988] do, but instead of only projecting $\hat{D}$ on the cone of negative semi definite matrices, without any rank constraint, we project onto $\kappa(r)$ using Algorithm 2. The second, projection is then onto the the space of hollow matrices. Alternating these projections gives us an iterative method to compute $D_t$.

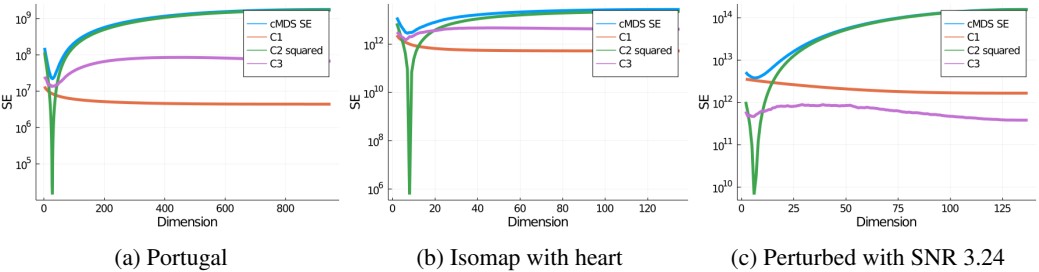

|     |     |     |
| --- | --- | --- |
| (a) Portugal | (b) Isomap with heart | (c) Perturbed with SNR 3.24 |

Figure 1: Plots showing the cMDS error as well as the three terms that we decompose the error into. For the perturbed EDM input, this is error with respect to the perturbed EDM.

## 4 Experiments

In this section, we do two things. First, we empirically verify all of the theoretical claims. Second, we show that on the downstream task of classification, if we use cMDS to embed the data, then as the

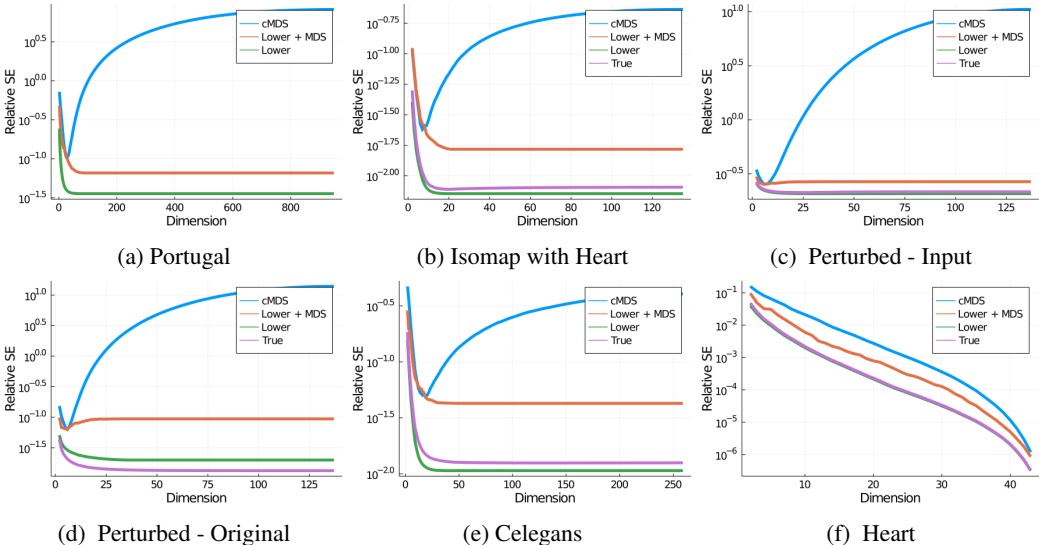

Figure 2: Plots showing the relative squared error of the solutions with respect to the input matrix. For the perturbed EDM input, we show the relative squared error with respect to the original EDM (figure (d)) and the perturbed EDM (figure (c)). For the Portugal data set, we couldn't compute the true solution due to computational restraints.

embedding dimension increases, the classification accuracy gets worse. Thus, suggesting that the embedding quality of cMDs degrades.

**Verifying theoretical claims.** To do this, we look at three different types of metrics. First, are metrics that come from graphs and for these we use Celegans Rossi and Ahmed [2015] and Portugal Rozemberczki et al. [2019] datasets. Second, are metrics obtained in an intermediate step of Isomap. Third, are perturbations of Euclidean metrics. For both of these metrics we use the heart dataset Detrano et al. [1989]. In all cases, we show that the classical MDS algorithm does not perform as previously expected; instead, it matches our new error analysis. In each case, we demonstrate that our new algorithm Lower + cMDS outperforms cMDS in relation to SSTRESS.

**Results.** First, let us see what happens when we perturb an EDM. Here we consider two different measures. Let $D$ be the original input, and $D_p$ be the perturbed. Then we measure

$$\frac{\|D_p - D_{\mathrm{cmds}}\|_F^2}{\|D_p\|_F^2} \quad \text{and} \quad \frac{\|D - D_{\mathrm{cmds}}\|_F^2}{\|D\|_F^2}.$$

As we can see from Figure 2c and 2d, as the embedding dimension increases both quantities eventually increase. The increase in the relative SSTRESS is as we predicted theoretically. The increase in the second quantity suggests that cMDS does not do a good job of denoising either. This suggests that the cMDS objective is not a good one.

Next, we see how cMDS does on graph datasets and with Isomap on Euclidean data. Here we plot the relative squared error ($\|D - D_{\mathrm{cmds}}\|_F^2 / \|D\|_F^2$). Figure 2 shows that, as predicted, as the embedding dimension increases, the cMDS error eventually increases as well. For both the Portugal dataset alone and with Isomap on the heart dataset, this error eventually becomes worse than the error when embedding into two dimensions! As we can see from Figure 1, this increase is **exactly** due to the $C_2^2$ term. Also, as heuristically predicted, the $C_3$ term is roughly constant.

Let us see how the true SSTRESS solution and new approximation algorithm perform. We look at the relative squared error again. First, Figure 2 shows us that our lower bound tracks the true SSTRESS solution extremely closely. We can also see from Figure 2d that the true SSTRESS solution and the Lower + cMDS solution are closer to the original EDM compared to the perturbed matrix. Thus, they are better at denoising than cMDS. Finally, we see that our new algorithm Lower + cMDS performs better than just cMDS and, in most cases, fixes the issue of the SSTRESS error increasing with dimension. We also see that $\|D_{\mathrm{lcmds}} - D\|_F^2 - \|D_t - D\|_F^2$ is also roughly constant.

We point out cMDS and Lower+cMDS have comparable running times as they can respectively be computed with one or two spectral decompositions. Computing $D_t$ requires computing a spectral decomposition in every round, with larger datasets requiring hundreds of rounds till they appear to converge. This is a significant advantage of Lower+cMDS over True, and also the reason we were not able to compute True for the Portugal dataset above, and the classifications experiments below.

**Classification.** For the classification tasks, we switch to more standard benchmark datatsets: MNIST, Fashion MNIST, and CIFAR10. If we treat each pixel as a coordinate then these datasets are Euclidean and we cannot demonstrate the issue with cMDS. We obtain non-Euclidean metrics in three ways. First, we compute the Euclidean metric and then perturb it with a symmetric, hollow matrix whose off diagonal entries are drawn from a Gaussian. Second, we construct a $k$ nearest neighbor graph and then compute the all pair shortest path metric on this graph. This is the metric that Isomap tries to embed using cMDS. Third, we imagine each image has missing pixels and then compute the distance between any pair of images using the pixels that they have in common (as done in Balzano et al. [2010], Gilbert and Sonthalia [2018]). Thus, we have 9 different metrics to embed. For each dataset, we constructed all 3 metrics for the first 2000 images. We embedded each of the nine metrics into Euclidean space of dimensions from 1 to 1000 using cMDS and our Lower + cMDS algorithm. We do not embed by solving Equation 2 as this is computationally expensive. For each embedding, we then took the first 1000 images are training points and trained a feed-forward 3 layer neural network to do classification. Results with a nearest neighbor classifier are in the appendix. We then tested the network on the remaining 1000 points. We can see from Figure 3, that as the embedding dimension increases, the classification accuracy drops significantly when trained on the cMDS embeddings. On the other hand, when trained on the Lower + cMDS embeddings, the classification accuracy does not drop, or if it degrades, it degrades significantly less. Thus, showing that the increase in SSTRESS for cMDS on non Euclidean metrics, does result in a degradation of the quality of the embedding and that our new method fixes this issues to a large extent.

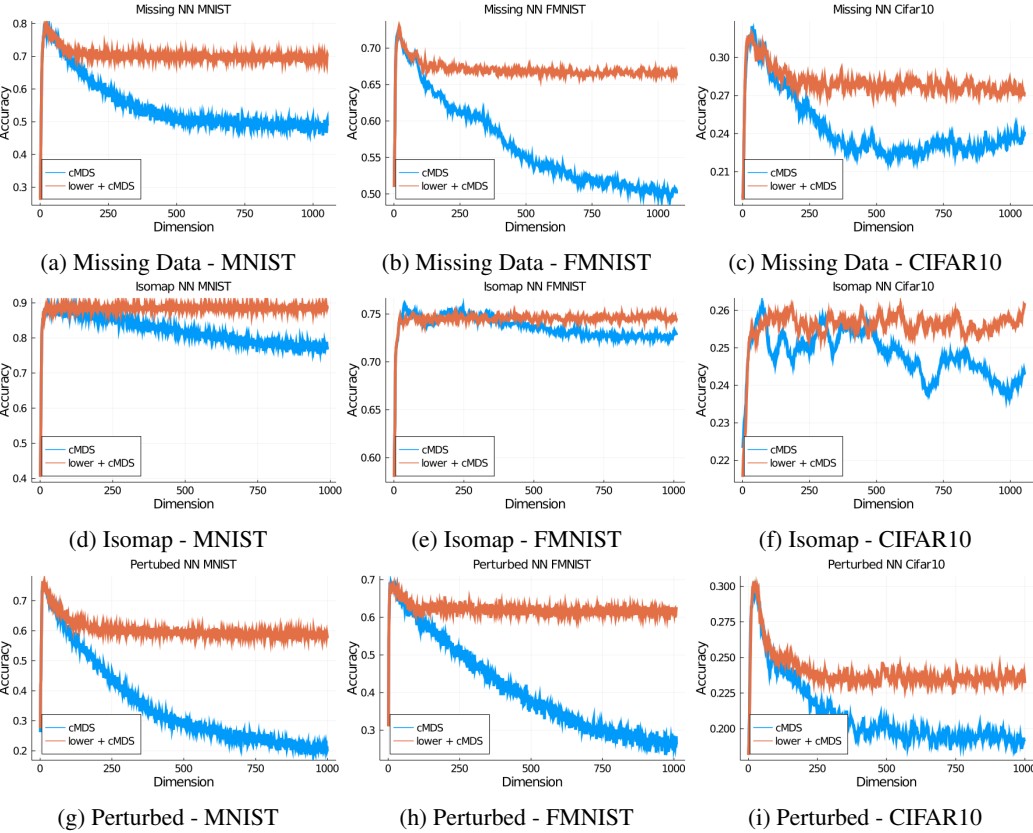

Figure 3: Plots showing the classification accuracy for a 3 layer neural network.

## Acknowledgements

Work on this paper by Gregory Van Buskirk and Benjamin Raichel was partially supported by NSF CAREER Award 1750780.

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
