# OpenReview forum: "How can classical multidimensional scaling go wrong?"
_NeurIPS.cc/2021/Conference — NeurIPS 2021 Poster_

### Official Review · Reviewer_rStB · 2021-07-07

**Rating:** 7
**Confidence:** 4

**Summary:**

The paper considers the classic embedding method known as multidimensional scaling and focus on the so called STRAIN  and SSTRESS objectives, both designed for embedding a given squared distance matrix into a lower dimensional space, often for visualization, but potentially also for reducing problem size for downstream tasks.

The idea is that the SSTRESS is the better objective to optimize as it captures how well the distances of the embedded data points approximate the input distances, but computationally hard, so maybe it is enough to just use the solution for the simpler STRAIN objective, that boils down to Singular Value Decomposition and that may work fine.

The paper analyzes the solution of STRAIN in terms of the more relevant SSTRESS objective, and shows a decomposition of the error of the STRAIN solution as measured with the STRESS objective and analyzes this objective to understand what happens.
The punchline is that if the input distance matrix is not Euclidian (no data set with these pairwise distances exist in any dimension), then the SSTRESS error of the embedding achieved by optimizing STRAIN can  actually increase with increased embedding dimension. A property that seems counterintuitive as a higher dimension should allow improved embeddings. The paper shows  experimentally that this phenomena actually occurs in practice as well and that the performance is nicely explained by the error decomposition proved and discussed in the paper.
In particular the experiments show that the classic Isomap embedding has this issue.
Furthermore, if these STRAIN embeddings are used for a simple downstream task like standard classification, the results become increasingly worse as the embedding size is increased.


**Limitations And Societal Impact:**

Yes

**Main Review:**

While the STRAIN objective may make most sense for Euclidian inputs and the failure is analyzed for non-euclidian, the experiments also show that just adding noise to the input distances may greatly worsen the results of using STRAIN based embeddings.  Based on the error decomposition the authors propose a way to fix the problem with the STRAIN objective, while maining the computational efficiency of the STRAIN method.
The fix improves the embedded distances, and improves the results for downstream tasks as the performance decrease significantly less, but it still seems there may better methods if a downstream classification task is the goal, and for me the experiments with the downstream task is more a proof of the failure of the STRAIN method to produce useful embeddings than that the new method is good solution for the dimensionality reduction problem.

In short, the paper shows that using STRAIN embeddings based for instance on IsoMap or other non-euclidean metrics to make a (semi) low dimensional embedding for instance for  downstream tasks is not good idea and should discourage practitioners from doing so, at least without proper thought. The error decomposition is computable and can be used to gauge into the potential issues with applying the STRESS optimization.


Technically the theoretical results seem sound and interesting and the analysis and result very relevant.
I think the same is true for the experiments shown in the paper.

A few things that may be improved/changed
- For the plots the y-axis is different in every plot which makes the pattern look identical on many experiments which cheated the viewer (me)  before I noticed.
- The claims of roughly constant of C3, C4, ... may be a slight exaggeration but the experiments shown they are clearly lower order so that may be okay. Actually C1 seems just as constant in the experiments.
- For very low dimensional embeddings it seems interesting things are happening and here it seems that STRAIN may still be useful at least on the data sets considered. Maybe a few comments on why that may be would be helpfull from a geometric or eigenvalue spectra kind of perspective.


Overall I think this line of research that explains the performance of well known algorithms is relevant and important.


**Time Spent Reviewing:**

6

---

> ### Author Response · Authors · 2021-08-06
> **Response**
>
> We thank the reviewer for the great summary of our paper!
>
> *"For the plots the y-axis is different in every plot which makes the pattern look identical on many experiments which cheated the viewer (me) before I noticed."*
>
> We do not mean to cheat anyone. Is this for figures 1,2,or 3 (or all of them)? The y axis for each subfigure in any given figure has the same quantity plotted on the y axis. However, the scales are different for the different subfigures due to the differences in the data.
>
> Note that in Figure 2(d) we are plotting $\||D_t - D\||$ and $\||D_l - D\||$ where $D$ is the original unperturbed metric. However $D_t$ and $D_l$ are computed by the various algorithms with $D+N$ as input where $N$ is a noise matrix. We plot $\||D_t - D - N \||$ and $\||D_l - D - N\||$ in Figure 2(c).
>
> We will work in improving the clarity of these figures.
>
>
> *"The claims of roughly constant of C3, C4, ... may be a slight exaggeration but the experiments shown they are clearly lower order so that may be okay. Actually C1 seems just as constant in the experiments."*
>
> $C_3$ and $C_4$ being constant are definitely heuristic estimates and not a formal claim. We will rephrase this is to say that they are small compared to $C_2$.
>
> $C_1$ is just the sum of eigenvalues squared and is easy to understand, whereas $C_3$ and $C_4$ are more opaque and hence we have our heuristics. Note that $C_1$ is eventually equal to the sum of the squares of the negative eigenvalues, and we agree about its apparent behavior.
>
> *"For very low dimensional embeddings it seems interesting things are happening and here it seems that STRAIN may still be useful at least on the data sets considered. Maybe a few comments on why that may be would be helpfull from a geometric or eigenvalue spectra kind of perspective."*
>
> Here we think the best performance happens when the $C_2$ term is smallest. This happens when the sum of the eigenvalues that we do not use is closest to 0.
>
> One possible informal explanation for this is that we can think of the negative eigenvalues as noise (for the purpose of Euclidean embeddings), and because our distance matrices have trace zero, we imagine that there are small positive eigenvalues that should be deemed noise as well.
>
> So when $C_2$ is close to 0, we are somehow ignoring the appropriate eigenvalues and the STRAIN method is picking the correct eigenvalues.

---

> > ### Comment · Reviewer_rStB · 2021-08-27
> > **Update**
> >
> > I thank the authors the response. I did not mean to imply any cheating, and i apologize if my comment could be understood this way.
> > After reading the other reviews and responses i still think this is a very nice paper and result.

---

### Official Review · Reviewer_f18Z · 2021-07-13

**Rating:** 6
**Confidence:** 3

**Summary:**

The authors study the problem of multidimensional scaling (MDS) for embedding data into Euclidean space and perform a theoretical error analysis of the classical muldimensional scaling (cMDS) algorithm. Their main result decomposes the error of cMDS into 3 components, which they use to explain the behaviour and degradation of the cMDS error with respect to the embedding dimension. Then, they propose an algorithm, based on their theoretical findings, and show that it outperforms cMDS on standard image classification datasets.

**Limitations And Societal Impact:**

The paper would benefit from a conclusion section summarizing the main contribution of the authors and discussing the strengths and limitations of the study.

The authors could also discuss potentially negative societal impacts if the method is applied for discriminative purposes on image datasets.

**Main Review:**

## Originality

The authors mention in the abstract that a theoretical analysis for cMDS on non-Euclidean metrics is lacking. Can the authors mention the error analysis on Euclidean metrics and detail how their approach differs from the literature, as well as the difference between their algorithm and the ones mentioned in l.84.

## Quality

Section 3 would benefit from a toy example where the 3 components of the errors are plotted with respect to the embedding dimension to validate the authors interpretation and the different weights between C1-C3. Maybe refer to Fig. 1 and use it to illustrate the discussion.

l. 102: The authors claim that they provide an "efficiently computable algorithm" but I do not see any argument in the paper supporting this claim such as a complexity analysis, comparison with other approaches, or benchmarks of computational time. l.216-217 mention that the major computational step is a spectral decomposition, is it affordable in large dimensions ? What are the complexity for the type of matrices considered (is it sparse, dense, structured matrices) ?

l. 231: The authors mention that solving eq. (3) is much more computationally intensive than their approach, can they explain the reasons as well as an order of magnitude for the difference (i.e. is it 10, 100 times more expensive) ? Similarly, they mention another approach by Qi and Yuan but do not do any comparison with them.

Figure 1: Why is the squared error so large ? Should it also not be normalized by ||D||_F^2 ?

l.84, the authors refer to several papers which aim to address the robustness with respect to noisy data but do not compare their approach with the literature in the experiments of section 4.

In the classification paragraph in section 4, why are the authors using a simple 3 layer feed-forward neural network ? Image classification taks are traditionally done using a convolutional neural network.

## Clarity

The paper is not very well written as it lacks clarity in several places, mostly in section 4.

What is the motivation for using the characterization by Hayden and Wells ?

I do not think that the presentation of Lemma 1-5 in the main text really supports the discussion. When describing the idea of the proof of Thm. 1, the authors could say that each of the three terms are identified to C1, C2^2 and C3, and point to the Supplementary Material.
On the contrary, the interpretation of these terms is central and would deserve a paragraph title for each to be clearly identified.

In Section 4, the discussion around Figures 1 and 2 is very confusing: the captions do not contain enough informations to understand the figures alone (they should at least define the terms in the legend). The organization of the figures is also confusing and it may be preferable to either discuss the decomposition of the error earlier in section 3 or organize the figures according to the datasets (e.g. for the Portugal one, having one panel for the decomposition and a second for the algorithm performance).
In Figure 2: the contribution would be clearer is the authors clearly identify their algorithm in the legend of Fig. 2. I do not understand as well what is the "Lower" curve and why the associated error is below the true solution.

The classification paragraph in section 4, the associated supplementary material, and the caption of Figure 3 lack informations to really understand and reproduce the experiment of the authors.

Minor comments:
- l. 120 "if and only if"
- l. 121: Please define the space in which $\mathbf{1}$ and $s$ live.
- Definition 2: I found it confusing that the notations do not exactly match the one given in eq. (5).
- l.161: Recall that $S\circ S$ denotes the Hadamard product
- l.174: "we can see that C1"
- l. 233: "Here we use"

## Significance

The error analysis of the authors explains the behaviour of the cMDS error, which is validated experimentally in Fig. 1. In Section 4, the authors show that their new algorithm outperform cMDS and does not degrade with respect to the embedding dimension.

**Time Spent Reviewing:**

4 hours

---

> ### Author Response · Authors · 2021-08-06
> **Response to Concerns**
>
> We thank the reviewer for their comments.
>
> # Originality Comments
>
> *"The authors mention in the abstract that a theoretical analysis for cMDS on non-Euclidean metrics is lacking. Can the authors mention the error analysis on Euclidean metrics and detail how their approach differs from the literature, as well as the difference between their algorithm and the ones mentioned in l.84."*
>
> To understand the analysis for cMDS thoroughly, we must be careful about which error function we are talking about. For the STRAIN metric, there is complete analysis for both the Euclidean and non-Euclidean metrics in terms of the unused eigenvalues of $-\frac{1}{2}VDV$. In fact, the error for the STRAIN metric is precisely the term $C_1$ in our analysis.
>
> If we look at the STRESS objective, then we see the additional two terms $C_2$ and $C_3$. Our formula works for both Euclidean and non-Euclidean metrics as discussed in Lines 177-178. All three of these terms can be seen to go 0 for Euclidean metrics as we increase the embedding dimension.
>
> # Quality Comments
>
> ## Computational Complexity
>
> Both cMDS and our method are $O(n^3)$. We will add a proposition detailing this.
>
> The only difference between our method and cMDS is that we shift the eigenvalues linearly. Computing this scaling can be done in time $O(n)$. The matrix multiplications are $QD$ or $VD$ where $Q$, $V$ are the identity matrix plus a rank 1 perturbation. Hence, the matrix multiplications can be performed as a sequence of matrix-vector multiplications. These matrix-vector multiplications are swift as well and can be done in time $O(n^2)$. There are two $O(n^3)$ matrix multiplications. The remaining step in the algorithm is the spectral decomposition which can be done in time $O(n^3)$.
>
> Qi and Yuan solve problem (3) and we use our own, different method to do so. To see the differences in the performance of the methods, we note that Qi and Yuan solve the problem for a 1740  by 1740 input matrix in 6678 seconds (albeit this is using resources from 2014). Our method for solving (3) is an iterative method that computes a spectral decomposition for each iteration. The method usually requires a few hundred iterations. Furthermore, we can compute spectral decompositions for 2000 by 2000 matrices in about 2 seconds on a laptop from 2017.
>
>
> *"Figure 1: Why is the squared error so large ? Should it also not be normalized by ||D||_F^2 ?"*
>
> The squared error can also be normalized. Doing so doesn't change the main takeaway of the experiments. $C_2$ increases and is many orders of magnitude bigger past a certain dimension.
>
> *"l.84, the authors refer to several papers which aim to address the robustness with respect to noisy data but do not compare their approach with the literature in the experiments of section 4."*
>
> **The goal of the paper is not to develop the best dimensionality reduction technique, but to understand the robustness of cMDS. For this, we provide an error decomposition, identify a problematic term, show that this has negative effects, and demonstrate that accounting for this term helps mitigate the negative effects.**
>
> *“In the classification paragraph ... a convolutional neural network.”*
>
> The network is trained on the embedded vectors. Even though the original vectors have the nice structure that CNNs exploit, the embedded vectors are just Euclidean vectors. Hence it is unclear if they have the structure of an image and so we used a feed forward neural network.
>
> # Clarity Comments
>
> *“What is the motivation for using the characterization by Hayden and Wells ?”*
>
> There is a strong relationship between cMDS and the eigenvalues of the matrix so we used the representation from Hayden and Wells.
>
> *“I do not think that the presentation of Lemma 1-5 ... a paragraph title for each to be clearly identified.”*
>
> We agree that not having the proofs in the main text reduces clarity. However due to the space constraints we had to move the proofs.
> The three paragraphs after the Lemmas do discuss where the terms came from and how they contribute to the error.
>
> *“In Section 4, … In Figure 2: the contribution would be clearer is the authors clearly identify their algorithm in the legend of Fig. 2. I do not understand as well what is the "Lower" curve and why the associated error is below the true solution.”*
>
> Lower is $D_l$ (equation 8) and it is what is returned by Algorithm 2 (lower), it lies below the true solution due to the analysis on lines 197 to 206.
>
> Note that in Figure 2(d) we are plotting $\||D_t - D\||$ and $\||D_l - D\||$ where $D$ is the original unperturbed metric. However $D_t$ and $D_l$ are computed by the various algorithms with $D+N$ as input where $N$ is a noise matrix. We plot $\||D_t - D - N \||$ and $\||D_l - D - N\||$ in Figure 2(c).
>
> *“The classification paragraph in section 4, … reproduce the experiment of the authors.”*
>
> We provide code as well for the experiments to enhance reproducibility.

---

> > ### Comment · Reviewer_f18Z · 2021-08-23
> > **Response to the authors**
> >
> > Thanks to the authors for the response.
> > The authors addressed most of my concerns and I have updated my score accordingly.

---

### Official Review · Reviewer_rAum · 2021-07-15

**Rating:** 7
**Confidence:** 3

**Summary:**

A new decomposition of the classical multidimensional scaling (cMDS) error term is generated and used to get additional insights of the method.
As I first came across the draft, I naively thought; “Well, cMDS is extremely well known, how can one innovate on such topic?”. Moreover, the departing justification stating cMDS does not excel in non-linear mappings wasn’t that great; I mean cMDS ASSUMES Euclidean space and linearity so it was no wonder that used out of its natural scope it may be suboptimal. But I am very glad to say that the authors proved my naivety wrong very quickly! There is a very clear contribution, the paper is strongly supported, and reading enjoyable.

STRENGTHS
+ The unusual “late comer”. Innovating in a method which is so well known is difficult.

WEAKNESSES
+ There are some unclear decisions and justifications but it is somewhat contradictory that I’m criticizing this when, I in the place of the authors, I would very likely have proceed equally… Notwithstanding, I’m indicating this below, in case a second thought can be given
+ There is an unconventional mixture of theoretical claims with experimental testing whereby some “theoretical proofs” are complemented with heuristics verified experimentally, but actually not theoretically proven.


**Limitations And Societal Impact:**

Not included, but limitations of cMDS are well known and I do not think any evaluation of the societal impact is needed for this research.

**Main Review:**

+ ln 68-71: These advantages are a bit simplistic (and yet I would claim them equally!). The 3rd one is particularly weak; the error term guides the optimization but it does NOT measure the quality of the solution. The same error value can represent wildly different solutions. A better measure of quality of the embedding is the pairing between embedded vs real distances (analogous to a QQ-plot). And of course, this is not the only one, there are other measures of the quality of a solution e.g. topological stability, etc.
+ ln 99-101: Ok but this seems a bit circumstantial. Can we guarantee that such relation holds for other tasks? Again, the quality of the embedding cannot be measure from the error.
+ Perhaps I’m wrong on this but in Definition 1, I think a limit to d is missing, otherwise as long as d=n+1 there will always be a valid solution as one can form the n simplex, and hence every matrix D will eventually be EDM.
+ ln 216-217: Sorry but this is NOT an analysis of complexity. You are very likely right, but unless you prove it formally or at least empricially, I would suggest to remove claims about running times.
+ “unconventional mixtures of theoretical claims with experimental testing”;
  * ln 228: I am unused to accept an heuristic as a part of a theoretical proof, unless the heuristic is itself theoretically proven (at which point, of course it ends being an heuristic).
  * ln 239: Well, something that is formally proven doesn’t need any additional experimentation, does it?
+ There is almost no discussion.
+ There is no conclusions section.

+ It is becoming more and more common to move the proofs to supplementary material, and the authors adhere to this practice. I disclose here that I very much dislike this practice, specially when the proofs are central to the research, and not a complement. Moreover, here the appendix are 7pgs!! If you truly needed 13+7=20pgs to communicate your research then perhaps a conference was not the right target for this research. But having said that, it is of course ultimately the authors decision, and I’m just being a grumpy old man! Please ignore my outburst, and proceed as you please.
+ I have been unable to inspect the code in detail as it seems to be in Julia (JLD) which is a language I do not master. Sorry.


**Time Spent Reviewing:**

6h

---

> ### Author Response · Authors · 2021-08-06
> **Response to Concerns**
>
> We thank the reviewer for their detailed comments.
>
> *“Perhaps I’m wrong on this but in Definition 1, I think a limit to d is missing, otherwise as long as d=n+1 there will always be a valid solution as one can form the n simplex, and hence every matrix D will eventually be EDM.”*
>
> We do not need the limit on $d$. Not every metric is eventually embeddable into Euclidean space. The Hayden and Wells characterization tells us that we have some conditions on the distance matrix.
>
> We can construct simple distances matrices such
>
> $$D^2 =\begin{bmatrix}  0 & 1 & 1 & 1 \\\\ 1 & 0 & 4 & 4 \\\\ 1 & 4 & 0 & 4 \\\\  1 & 4 & 4 & 0\end{bmatrix}$$ does not satisfy these conditions. But this is a valid metric.
> The eigenvalues of $-\frac{1}{2}VD^2V$ for this $D^2$ are $2,2,0,-0.25$. Hence due to the eigenvalue $-0.25$ this metric will not be embeddable in any dimensional Euclidean space!
>
> *"ln 216-217: Sorry but this is NOT an analysis of complexity. You are very likely right, but unless you prove it formally or at least empricially, I would suggest to remove claims about running times."*
>
> We agree. As we mention in the response to Reviewer f18z we will add a formal proposition. That response has a more detailed analysis of the running time.
>
>
> *“unconventional mixtures of theoretical claims with experimental testing”;
> ln 228: I am unused to accept an heuristic as a part of a theoretical proof, unless the heuristic is itself theoretically proven (at which point, of course it ends being an heuristic).
> ln 239: Well, something that is formally proven doesn’t need any additional experimentation, does it?”*
>
> We think the confusion here is due to the presentation of Lemma 6! Lemma 6 does not include line 225 and the equations following line 225. The discussion follows immediately after the statement of Lemma 6; that is, from line 255 onwards, we discuss only heuristics.
>
> We will clarify the phrasing here to add that we check our heuristics and to demonstrate the difference in magnitude between the $C_2$ terms and the other terms.

---

> > ### Comment · Reviewer_rAum · 2021-08-25
> > **Negative eigenvalues and embeddability in metrics spaces**
> >
> > Thanks for the answers and in particular for the example regarding about the non-embeddable distance matrix. This is however not fully clear to me. While it may be the case that degenerated distance matrices (those with negative eigenvalues) may not be "directly" embeddable, but there exist several bending procedures that can transform a matrix with negative eigenvalues in another one where all its eigenvalues are positive e.g. [Schaffer2010]. Whether any bending can preserve the metric properties I do not know, but my point is that having negative eigenvalues per se may not be enough to prevent embeddability, only make it more difficult. Anyway, I still remain uncertain on this on my side, so I take the word of the authors for it.

---

> > > ### Author Response · Authors · 2021-08-25
> > > **Proof for the Non-Embeddability**
> > >
> > > Here is a proof showing that the example from the previous comment is not an EDM as defined by definition 1 (i.e., without a limit on the dimensionality). This metric is an example for something that is not an EDM in [2] that is left as an exercise.
> > >
> > > Let $D = \begin{bmatrix} 0 & 1 & 1 & 1 \\\\ 1 & 0 & 2 & 2 \\\\ 1 & 2 & 0 & 2 \\\\ 1 & 2 & 2 & 0 \end{bmatrix}$. This is the matrix from the previous example. ($D^2$ from previous comment was the matrix with the entries squared)
> > >
> > > Assume for the sake of contradiction that there exists points $x_1, x_2, x_3, x_4$ that realize this metric. That is, $|| x_i - x_j || = D_{ij}$
> > >
> > > Since Euclidean metrics are translationally invariant, i.e., for any vector $v$, we have that $||(x_i-v) - (x_k-v)|| = ||x_i - x_j||$, we can translate $x_i$s as follows $\hat{x_i} = x_i - x_1$.
> > >
> > > After this translation, we see that $\hat{x_1} = 0$. Again since Euclidean metrics are rotationally invariant, we can rotate the $\hat{x_i}$ to get $\tilde{x_i}$ such that $\tilde{x_2} = [1\ 0\ \ldots\ 0]^T$. Since the origin doesn't move when we rotate, we get that $\tilde{x_1} = 0$.
> > >
> > > Finally, we know that $||\tilde{x_2} - \tilde{x_3}|| = ||\tilde{x_2} - \tilde{x_4}|| = 2$. Thus, $\tilde{x_3},\tilde{x_4}$ live on a the surface of a sphere of radius 2 centered at $\tilde{x_2} = [1\ 0\ \ldots\ 0]^T$.
> > >
> > > Similarly, we know that $||\tilde{x_1} - \tilde{x_3}|| = ||\tilde{x_1} - \tilde{x_4}|| = 1$. Thus, $\tilde{x_3},\tilde{x_4}$ live on a the surface of a sphere of radius 1 centered at $\tilde{x_1} = 0$.
> > >
> > > Let $[v_1\ v_2\ \ldots\ v_k]$ be a vector in the intersection of these two spheres. Then $\sum_{i=1}^k v_i^2 = 1$ and $4 = (v_1-1)^2 + \sum_{i=2}^k v_i^2$.
> > >
> > > Substituting the first into the second, we see that $2 = -2v_1 \Rightarrow v_1 = -1$. Plugging this back into the first, we see that $\sum_{i=2}^k v_i^2 = 0$. Thus, for $i \ge 2$, $v_i = 0$.
> > >
> > > Thus, the intersection of the two spheres is the unique point $[-1\ 0\ \ldots\ 0]^T$. However, by assumptions, there exist two distinct points $\tilde{x_3}, \tilde{x_4}$ that lie in this intersection. Hence we have a contradiction.
> > >
> > > Thus, we couldn't have had points $x_1, x_2, x_3, x_4$ that realized the metric $D$. Note we had no assumptions on the dimensionality of $x_1, x_2, x_3, x_4$.
> > >
> > > We refer the reviewer to the following for more discussions on the non-embeddability of metrics in Euclidean space with the $\ell_2$ metric.
> > >
> > > 1) Nathan Linial, Eran London, and Yuri Rabinovich. The geometry of graphs and some of its algorithmic applications. Combinatorica, 15(2):215–245, Jun 1995.
> > > 2) Lecture notes on metric embeddings by Jiri Matousek

---

> > > > ### Comment · Reviewer_rAum · 2021-08-25
> > > > **Proof for the Non-Embeddability**
> > > >
> > > > Outstanding! Thanks! Will definitively read those refs!.

---

### Official Review · Reviewer_BDxF · 2021-07-16

**Rating:** 6
**Confidence:** 3

**Summary:**

The work shows a scenario that the classical MDS algorithm fails. "When the derived matrix has a significant number of negative eigenvalues," the classical MDS algorithm tends to fail when the number of dimensions increases sufficiently large. Then the work designs an algorithm with improved performance bound and shows the potential on practical applications.

**Main Review:**

Pros:

1. The writing is good and gives clear explanations and detailed background for derivations, which are beneficial for both experienced and non-experienced readers.

2. The analysis results seems reasonable to me.

3. The resulting algorithm applies the Dykstra's method to iteratively projection onto the space of hollow matrices and achieve a fast solution procedure.

Concerns:

1. In MDS, typically people use a small number of dimensions for embedding. So the scenario of failures described in the paper seems rare, which may limit the practical significance of the results in the work.

2. In literature, there are known issues on the classical MDS. For example from https://en.wikipedia.org/wiki/Isomap, "following the connection between the classical scaling and PCA, metric MDS can be interpreted as kernel PCA." ... "However, the kernel matrix K is not always Positive Semi-Definite. The main idea for kernel Isomap is to make this K as a Mercer kernel matrix (that is positive semidefinite) using a constant-shifting method, in order to relate it to kernel PCA such that the generalization property naturally emerges." Is there any relationship between the paper's finding and this known issue of non-PSDness? How to compare the proposed algorithm and the kernel PCA algorithm? It is preferred to see some discussion on this point in the rebuttal phase.


**Time Spent Reviewing:**

~4

---

> ### Author Response · Authors · 2021-08-06
> **Response to Concerns**
>
> We thank the reviewer for their comments.
>
> # Kernel PCA, ISOMAP and connections
>
> For kernel PCA, one seeks to factor a kernel matrix. From [1], the connection between the classical MDS (cMDS) method and kernel PCA arises from the fact that we compute the eigenvalue decomposition of $-\frac{VDV}{2}$ (i.e. $HKH$ in the notation of [1] or $K(D^2)$ in the notation of [2]) for cMDS.
>
> As the reviewer points out, when $-\frac{VDV}{2}$ is not positive semi definite, we have a problem! [2] provides a method by which we can shift the matrix to make it positive semi definite.
>
> That is not, however, the goal of our work. The goal of our paper is to analyze the effect of the lack of positive semi definiteness. To do so, we provide our error decomposition and show that due to the negative eigenvalues of $-\frac{VDV}{2}$, the SSTRESS of the cMDS increases as the dimension increases.
>
> **Hence the main goal of our paper is to quantify the effect of the kernel not being positive semi definite**
>
> [1]  Williams, Christopher K. I.. “On a Connection between Kernel PCA and Metric Multidimensional Scaling.” Machine Learning 46 (2004): 11-19.
>
> [2] Choi, H.; Choi, S.: 'Kernel Isomap', Electronics Letters, 2004, 40, (25), p. 1612-1613, DOI: 10.1049/el:20046791
>
> # Dimension for embedding
>
> While cMDS is usually used to embed into small dimensions. We do not know, prior to our work, when the quality of the embedding will start to decrease. For example, CIFAR10 is a 3072 dimensional data set and embedding into only a 200 dimensional space would seem to be relatively insufficient for such a data set. However, from our plots, we see that the quality of the embedding has already started to degrade at dimension 200.
>
> Our method lets us estimate when the quality will start to degrade (using the $C_2$ term) and so it can be used to determine what the best low dimensional dimension to embed into is.
>
> Our work might also indicate why people only use cMDS for low dimensional embeddings.

---

> > ### Comment · Reviewer_BDxF · 2021-08-29
> > **Response**
> >
> > Thanks for the reply.
> >
> > I read all reviews/rebuttals carefully and re-read your paper. I like the proof from theoretical viewpoint, while still remain rather uncertain by its practical implication.
> >
> > A small point, CIFAR10 is a set of images with 1024 pixels. Maybe better to treat them as 1024-dim instead of 3072-dim of RGB channels.

---

### Decision · Program_Chairs · 2021-09-27

**Decision:**

Accept (Poster)

**Comment:**

This seems to be a strong theoretical contribution to NeurIPS, although both some of the reviewers and myself suspect it is less relevant for practical usage.